

# Uncertainty of eddy covariance flux measurements over an urban area based on two towers

Leena Järvi [1,2], Üllar Rannik [1], Tom V. Kokkonen [1], Mona Kurppa [1], Ari Karppinen [3], Rostislav D. Kouznetsov [3,4], Pekka Rantala [1], Timo Vesala [1,5], and Curtis R. Wood [3]

[1]Institute for Atmospheric and Earth System Research (INAR) / Physics, Faculty of Science, P.O. Box 68, 00014 University of Helsinki, Finland
[2]Helsinki Institute of Sustainability Science, Faculty of Science, P.O. Box 68, 00014 University of Helsinki, Finland
[3]Finnish Meteorological Institute, P.O. Box 503, 00101 Helsinki, Finland
[4]A. M. Obukhov Institute of Atmospheric Physics, 119017 Moscow, Russia
[5]INAR / Forest Sciences, Faculty of Agriculture and Forestry, P.O. Box 27, 00014 University of Helsinki, Finland

**Correspondence:** Leena Järvi (leena.jarvi@helsinki.fi)

**Abstract.** The eddy-covariance (EC) technique is the most direct method to measure the exchange between the surface and the atmosphere in different ecosystems. Thus, it is commonly used to get information on air pollutant and greenhouse gas emissions, and on turbulent heat transfer. Often one ecosystem is monitored using only a single EC measurement station bringing uncertainties to the ecosystem-level flux values. Furthermore, in urban ecosystems we are often compromised to

conduct the single-point measurements in non-ideal locations such as close to buildings and/or in the roughness sublayer bringing further complications to data analysis and flux estimations. In order to tackle the question of how representative a single EC measurement point in an urban area can be, two identical EC systems – measuring momentum, sensible and latent heat, and carbon dioxide fluxes – were installed on each side of the same building structure in central Helsinki, Finland, for July 2013–September 2015. The main interests were to understand what is the sensitivity of the vertical fluxes on the single

measurement point and to estimate the systematic uncertainty on annual cumulative values due to missing data if certain, relatively wide, flow-distorted wind sectors are disregarded.

The momentum and measured scalar fluxes respond very differently to the distortion caused by the building structure. The momentum flux is the most sensitive to the measurement location whereas scalar fluxes are less impacted by the measurement structures. The flow distortion areas of the two EC systems (40–150° and 230–340°) are best detected from the mean-wind-

normalised turbulent kinetic energy and outside these areas, the random uncertainties of the two systems are between 10 and 40 %. Different gap-filling methods to yield annual cumulative fluxes show how using data from a single EC measurement point can cause up to 12 % underestimation in the cumulative carbon fluxes when compared to combined data from the two systems. Combining data from two EC systems increases also data coverage from 45 % to 69 %. For sensible and latent heat, the respective underestimations are up to 5–8 %. We also show how the commonly used data flagging criteria in natural

ecosystems, kurtosis and skewness, are not necessarily suitable to filter out data in a densely built urban environment. The results show how the single measurement system can be used to derive representative flux values for central Helsinki but the addition of second system to other side of the building structure decreases the uncertainties. The same results can be assumed



to apply in similar dense city locations where no large directional deviations in the source area are seen. In general, the obtained results will aid the scientific community by providing information about the sensitivity of EC measurements and their quality flagging in urban areas.

*Copyright statement.* TEXT

## 1   Introduction

It is recommended that surface fluxes measured using the eddy-covariance (EC) technique are done in the inertial sublayer and free from obstructions (Roth, 2000). These assumptions are often easy to meet over natural surfaces–but can be challenging for EC systems above cities. Often the EC measurements are made within or in the vicinity of the roughness sublayer, the adjacent layer to the surface with height of 2–5 times the mean building height (Raupach et al., 1991). In this layer, turbulence is not homogeneous but rather varies greatly in space, and the Monin-Obukhov similarity theory (MOST) is not anymore strictly valid. Despite the non-ideal conditions, EC measurements from urban areas are needed for the purposes of wind engineering, understanding the urban surface-atmosphere interactions, in the estimation of urban carbon budgets (Christen et al., 2011; Nordbo et al., 2012a) and on improving the description of urban areas in numerical weather and air quality predictions via the measured turbulent fluxes of heat (Grimmond et al., 2010; Karsisto et al., 2015; Demuzere et al., 2017). In order for the urban EC systems to meet the requirements of the technique, we are often forced to conduct the measurements on top of buildings or other platforms such as telecommunication towers (Wood et al., 2010; Liu et al., 2012; Brümmer et al., 2013; Nordbo et al., 2013; Keogh et al., 2012; Ao et al., 2016) instead of narrow lattice masts which would minimise the effect of the structure itself on the EC measurements. Thus strictly speaking, the measurements are not necessarily made completely free of the impact of roughness elements even if the measurement height would be at sufficient height above the surrounding roughness elements. The interaction between the EC measurements and the measurement platform itself causes challenges for obtaining high quality EC datasets and special attention should be paid on the effect of the so-called flow distortion area on the measurements (Barlow et al., 2011). Urban EC measurements have furthermore raised the need for local scaling of mean turbulent properties with minor deviations from inertial-sublayer scaling (Rotach, 1993; Roth, 2000; Vesala et al., 2008; Wood et al., 2010) and corrections for local-scale anthropogenic sources (Kotthaus and Grimmond, 2012).

The basic quality screening of a single sensor in measuring vertical fluxes can be performed based on the vertical deflection angles and expected turbulence, and sometimes even by simply disregarding whole (flow-distorted) wind sectors. It is not however ideal if we have to reject large amount of data. For cumulative emission estimates, the flux data need to be gap-filled – but in urban areas this is more complex than in vegetated environments due to the large amount of explanatory variables and the high spatial variability of the sources and sinks (Menzer et al., 2015). The used gap-filling methods in urban EC flux datasets vary from simple look-up tables to artificial neural networks (Schmidt et al., 2008; Kordowski and Kuttler, 2010; Christen et al., 2011; Järvi et al., 2012), but the more complex and time-demanding solutions might not always be considerably



better than the more simple ones. Järvi et al. (2012) found only 4% difference in cumulative carbon dioxide ($CO_2$) fluxes when utilising median diurnal cycles and neural networks in filling data gaps at a semi-urban site in Helsinki. Either way, statistical gap-filling techniques can be biased if certain wind directions are compromised above heterogeneous surfaces and therefore single-point EC measurements might not give realistic cumulative flux values. The same applies to the representativeness of a

single measurement point for a studied ecosystem in general. Already at forested sites, which are generally considered to be easier for EC measurements than urban areas, the uncertainties in $CO_2$ flux originating from a single measurement point have been reported to be 6 % (Hollinger et al., 2012). In the past, simultaneous observations from more than one EC station have been used to estimate uncertainties in EC-measured fluxes above vegetated surfaces (Kessomkiat et al., 2010; Peltola et al., 2015; Post et al., 2015) but in urban areas no estimations have been derived from direct EC measurements with more than one

measurement system at same level.

The aim of this work is twofold. Firstly, we want to examine the sensitivity of a single-point EC system in measuring the vertical fluxes of momentum, sensible and latent heat, and carbon dioxide in a highly dense urban area. Secondly, we want to understand what is the implication of the non-ideal measurement location and resulting data removal on the calculation of cumulative fluxes, which are important for emission-inventory comparison and planning of neighbourhoods. These two aims

will be examined with the aid of two identical EC measurement systems located on the opposite sides of a bluff-body tower in the centre of Helsinki.

## 2   Methods

### 2.1   Measurement site and instrumentation

The measurements were conducted in central Helsinki (Figure **??**) where two identical EC setups were installed on top of a

hotel building (Figure 2) at height ($z$) of 60 m above the ground for July 2013 until September 2015. Within 1 km radius of the measurement location, 37 % of the surface is covered with buildings, 41 % with paved surfaces leaving only 22 % of the surface covered with vegetation (Nordbo et al., 2015). The surrounding buildings are fairly uniform with a mean height of 24 m, displacement height of 15 m and aerodynamic roughness length of 1.4 m (Nordbo et al., 2013). However one notable exception is the Hotel Torni building itself: its main building is up to 15 m above the ground level, the tower up to 58 m and upper masonry

extending up to 66 m. The two EC systems (EC1 and EC2) were mounted on the opposite sides of an upper masonry on a 2.3 m high measurement masts (Figure 2). Thus, the systems are located at 60.3 m which is 2.5 times the mean building height and therefore they should be above the roughness sublayer and blending height where local-scale surface sources and sinks have aggregated together (Raupach et al., 1991). The centre of Helsinki is located on a peninsula, but previous analyses on the source area of EC1 system have shown the flux footprint to lie above the city and not the sea (Kurppa et al., 2015; Auvinen

et al., 2017). The downside of the measurement location is that the upper masonry disturbs the flow and we choose to neglect data for certain wind directions based on quality considerations. Based on the mean-wind-normalised turbulent kinetic energy (TKE), the areas are approximated to be 40–150° and 230–340° for EC1 and EC2, respectively (Figure 3).



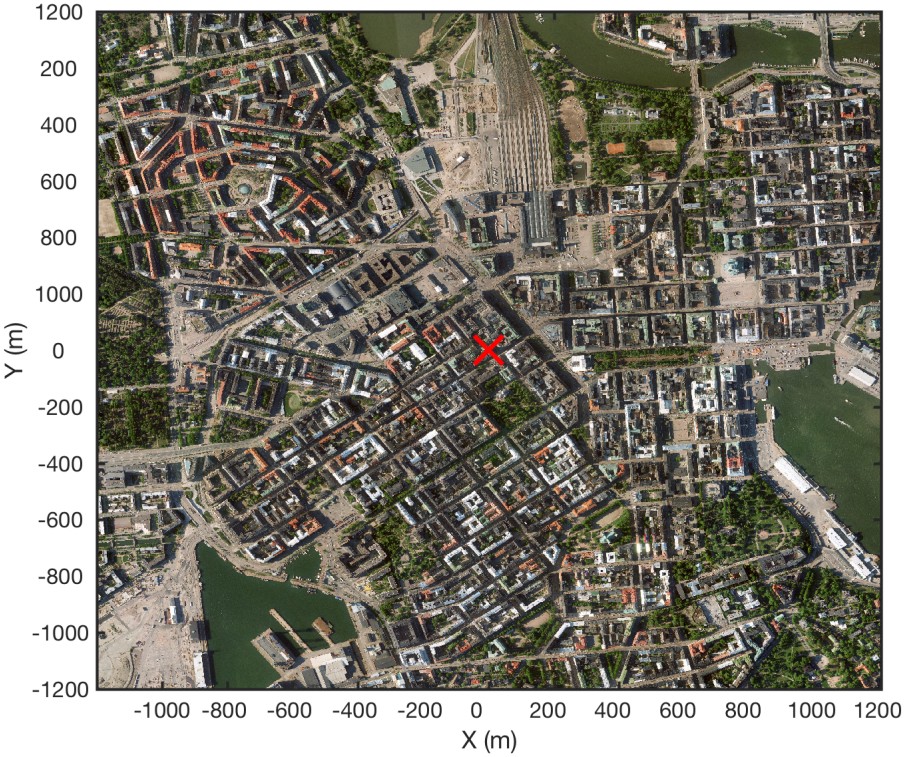

**Figure 1.** Aerial image of central Helsinki (Kaupunkimittausosasto, Helsinki, 2011). Hotel Torni is marked with red cross.

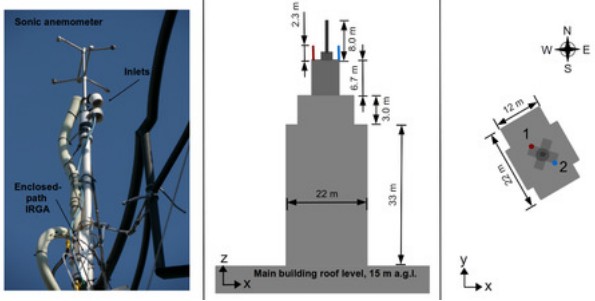

**Figure 2.** Left: a photo of one EC installation. Middle: a side view of the tower. Right: a plan view. See Nordbo et al. (2013) for more details.

Each system comprised of a 3D ultrasonic anemometer measuring the sonic temperature and 3D orthogonal wind speeds (USA-1, Metek GmbH, Germany), and an infrared gas analyzer (LI-7200, LI-COR Biosciences, Lincoln, NE, USA) giving concentrations of water vapour and $CO_2$. The air inlets were positioned 0.15 m below the anemometer centre and air was drawn





through a 1 m long stainless steel tube (with inner diameter of 0.04 m) to the gas analyser. The flow rates were $10 \, \text{l} \, \text{min}^{-1}$. Tubes were heated with a power of $9 \, \text{W} \, \text{m}^{-1}$ to avoid condensation of water vapour on their walls. The raw EC data were sampled with a frequency of 10 Hz, from which the 30-min flux values were calculated using commonly accepted procedures (Nordbo et al., 2012b). The fluxes were determined using the maximum covariance technique where the window mean and

width for the lag time were identical for the two systems (0–1.2 s for $CO_2$ and 0–7 s for $H_2O$). Before the calculation of the fluxes, data were despiked and linearly detrended. The high-response losses resulting from the tube attenuation were corrected with the aid of measured temperature cospectra yielding $CO_2$ response time of 0.11 s for EC1 and 0.14 s for EC2, respectively. Wind coming from the flow distortion areas removed 27 % of the EC1 data and 38 % of the EC2 data. The larger fraction with EC2 is due to the prevailing wind direction from South–West.

## 2.2   Data analysis

In order to understand possible differences between the two measurement setups, several variables and statistics describing turbulence characteristics will be evaluated. Stationarity ($FS$), skewness ($SK$) and kurtosis ($K$) are common variables used to examine the quality of EC data with the first providing information about the stationarity of the flux measurements and the latter two about the form of the probability function of the measured concentration, temperature or wind speed (Vickers

and Mahrt, 1997). Stationarity is calculated by dividing each 30-min flux period into six sub-sets for which the flux values are separately calculated and their mean furthermore compared with the 30-min flux values. Typically, with differences below 30%, data are considered to be high quality and differences below 60% still suitable for general data analysis. In this study, the strict limit of 30% will be used. $SK$ describes the asymmetry of the probability function of a variable and it is calculated from

$$SK = \frac{\overline{(x'^3)}}{\sigma_x^3}, \tag{1}$$

where $x$ is a velocity or scalar variable, overbar indicates the 30-min time average, prime the deviation from the mean of the variable and $\sigma_x$ is its standard deviation. $SK$ between –2 and +2 is commonly considered to be good quality EC data. $K$ is a measure of sharpness of the probability function i.e. its high values indicate peaks in the data. It is calculated from

$$K = \frac{\overline{(x'^4)}}{\sigma_x^4}. \tag{2}$$

$K$ between 1 and 8 is considered as reasonable quality data.

The relative random uncertainty (RRE) of the flux measurements is calculated as the square root of the random flux error ($\sigma_F^2$) relative to the suqare of the absolute value of the flux according to Lenschow et al. (1994).

$$\text{RRE} = \frac{\sigma_F(\Gamma)}{|F|} = \frac{2\Gamma_f}{\Gamma} \frac{1 + r_{ws}^2}{r_{ws}^2}, \tag{3}$$

where $\Gamma$ is the averaging period (30 min), $\Gamma_f$ the integral time-scale and $r_{ws}$ the correlation coefficient between the time series.

The turbulent kinetic energy (TKE) is obtained from

$$\text{TKE} = 0.5(\overline{u'^2} + \overline{v'^2} + \overline{w'^2}). \tag{4}$$





The turbulent transfer efficiencies for momentum and heat fluxes are calculated from

$$|r_{uw}| = \frac{\overline{u'w'}}{\sigma_u \sigma_w}, \tag{5}$$

and

$$|r_{wT}| = \frac{\overline{w'T'}}{\sigma_w \sigma_T}. \tag{6}$$

The power and co-spectra of momentum ($\tau$), sensible heat ($H$) and carbon dioxide ($F_c$) fluxes are calculated using fast Fourier transforms for 60-min periods ($2^{15}$ points). Spectra are divided into 76 logarithmically even-spaced bins. The frequency dependent atmospheric spectra

$$\frac{f S_x(f)}{\text{var}(x)} \tag{7}$$

and co-spectra

$$\frac{f S_{xw}(f)}{\text{cov}(x, w)}, \tag{8}$$

of the vertical flux and variable $x$ are plotted against the normalised frequency $n$:

$$n = \frac{f(z - d)}{U}, \tag{9}$$

where $f$ is the frequency (Hz) and $U$ is the mean wind speed.

## 3   Results

### 3.1   Turbulent transport and vertical fluxes

The flow distortion areas of both EC systems (no filtering based on $FS$, $SK$ and $K$) due to the upper masonry are clearly distinguishable from the vertical deflection angle ($\theta$), normalised TKE and turbulent transfer coefficients (Figure 3). Even though the two EC systems were to our best attempts designed to be identical and symmetrically located on the opposite side of the masonry, we observe quantitative asymmetry in the first and second moment statistics. The vertical deflection angle,
which sets $w = 0$ in the two-dimensional coordinate rotation ($\tan^{-1}(\overline{w}/U)$) and describes the distortion of the measurement structure on the measurements, experiences fluctuating behaviour in these areas indicating modified flow structure due to the building masonry (Figure 3a). Some of the deviation can be explained by variation in the surrounding topographies in the direction of flow distortion areas.

Outside the flow distortion areas, the angles vary between 5–18° with EC1 and between 2–15° with EC2 which are at
the same range as observed at BT Tower in London (Barlow et al., 2011). The normalised TKE at the flow distortion area measured with EC1 reaches 2.5 and with EC2 1.7 showing clearly the asymmetry in the areas. Both EC systems give a mean value of 0.34 for the normalised TKE outside the flow distortion areas indicating them measuring similar turbulence (Figure





3b). Furthermore, TKE is fairly uniform with wind direction despite the measurement location being considered to be complex from the point of view of micrometeorological measurements. Also the transfer efficiencies for heat are similar between the two systems with the values of 0.32 for EC1 and 0.29 for EC2 outside the flow distortion areas (Figure 3c). The transfer efficiencies of momentum are clearly different from that of heat and have largest deviations between the two systems (Figure 3d). The transfer coefficient for heat has a clear dip when the flow is disturbed whereas the momentum transfer coefficients follow a more complex pattern. This indicates the different effect of the measurement platform on the transport of momentum and heat with stronger effect on the first.

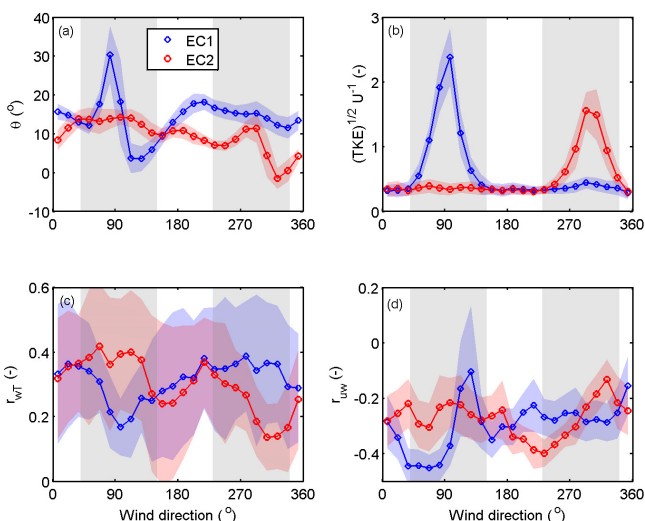

**Figure 3.** Wind direction dependence of (a) the vertical deflection angle ($\theta$), (b) normalised turbulent kinetic energy (TKE$^{1/2}$ U$^{-1}$) and turbulent transfer efficiencies (c) of heat ($r_{wT}$) and (d) momentum ($r_{uw}$) from EC1 and EC2 for July 2013 until September 2015. Only winds speeds $U > 1$ m s$^{-1}$ and for r$_{wT}$ $|H| > 10$ W m$^{-2}$ are taken into account. Lines and symbols represent the 15° bin averages and the patches $\pm$ 1 std. The disturbed wind directions (40–150° for EC1 and 230–340° for EC2) are marked with grey areas.

The asymmetry of the flow distortion areas is furthermore reflected in the vertical fluxes of momentum ($\tau$), sensible ($H$) and latent heat ($LE$), and CO$_2$ ($F_C$) (Figure 4). The strength of asymmetry varies with atmospheric stability and between variables indicating that purely prevailing meteorology cannot be responsible for the observed differences but rather the morphological effects play a role. Outside the flow distortion areas, differences between the two systems are small and depend on the studied flux. The best correlation between the two EC systems is seen in $H$ with the median of correlation coefficient (R$^2$) being 0.95, the slope of the linear least square regression (EC2 = Slope·EC1 + Intercept) being close to one and the intercept within $\pm 5$ W m$^{-2}$ (Figure 5). The maximum difference in the absolute values is 20 W m$^{-2}$ (Figure 4b) in unstable conditions. In the correlation of $\tau$, largest differences of all fluxes with a sinusoidal pattern as a function of wind direction are seen. The slope varies between 0.5–1.8 and the intercept is systematically below zero indicating lower momentum flux measured by the EC2



than EC1 (Figure 5a,b). Furthermore, the median $R^2$ is 0.85 (Figure 5c). The directional dependencies and correlations between the two systems in measuring $LE$ and $F_C$ follow a similar pattern indicating similarity between the two variables (Figures 4c,d and 5). For $LE$, the correlation statistics are however somewhat lower than for $F_C$. $LE$ has the correlation coefficient in the range of 0.6–0.9, the slope in the range of 0.7–1.0 and the intercept in the order of 10 W m$^{-2}$ with a greater flux measured

with EC2 than EC1. For $F_C$ the respective values are 0.8–0.9, 0.7–1.1 and 0–5 $\mu$mol m$^{-2}$ s$^{-1}$. The absolute differences yield -1.9 W m$^{-2}$ and -0.3 $\mu$mol m$^{-2}$ s$^{-1}$, respectively. The correlation statistics in our case are slightly poorer that observed over a a grassland in UK (Mccalmont et al., 2017), where $R^2$ scatter suggested sampling uncertainty between 5–7% when compared to our 10–20%.

The separation distance between the two EC systems is less than 10 m and thus they are expected to measure the same
source area outside the flow distortion areas. At the same time the observed differences cannot arise from the post-processing as fluxes were calculated and processed in a similar manner. Some of the difference can still originate from instrument drifting, but this would indicate non-directional dependence. As a result, the differences in the fluxes measured by the two systems very probably relate to the variation of the flux field caused by complex terrain. In past studies above vegetated ecosystems, the random uncertainty of flux measurements resulting from instrumental errors, heterogeneity of the surface and turbulence has
been determined using so-called two-tower approach (Hollinger and Richardson, 2005; Kessomkiat et al., 2010). Its assumption is that the two time series should be independent from each other and thus cannot be used in our case when the two systems are measuring the same footprint. We can however still calculate the relative random error (RRE) in order to get an understanding about the random uncertainties of our EC measurements. Of all studied vertical fluxes, the largest random uncertainties relate to $\tau$ (medians between 23–28%) and the lowest to $LE$ (medians between 16–20%) (Figure 6). For $\tau$ no systematic pattern
between daytime/night-time is seen whereas for the other fluxes, nocturnal uncertainties tend to be larger when also the scalar fluxes are small. For these also RRE from EC2 are slightly larger than from EC1 whereas for $\tau$ these are vice versa. The RREs are of the same order of magnitude as observed at the semi-urban site in Kumpula and above vegetated ecosystems. In these, however, the RRE associated with $\tau$ tends to be the lowest contrary to our study (Finkelstein and Sims, 2001; Billesbach, 2011; Nordbo et al., 2012b).

Both statistical variables RRE and $R^2$ should in principle provide the same information about random uncertainty. When RRE from the two systems are larger, $R^2$ should respectively be smaller. However, this is not observed and based on $R^2$ the fluxes can be ranked in increasing order $LE$, $F_C$, $\tau$ and $H$ both in day- and night-time (0.79, 0.82, 0.86, 0.92 and 0.66, 0.85, 0.88, 0.94). A possible explanation for this is that $R^2$ is calculated between the two EC systems and is impacted by systematic disturbances and the building masonry. Thus, RRE is considered to be more representative for flux random uncertainties.



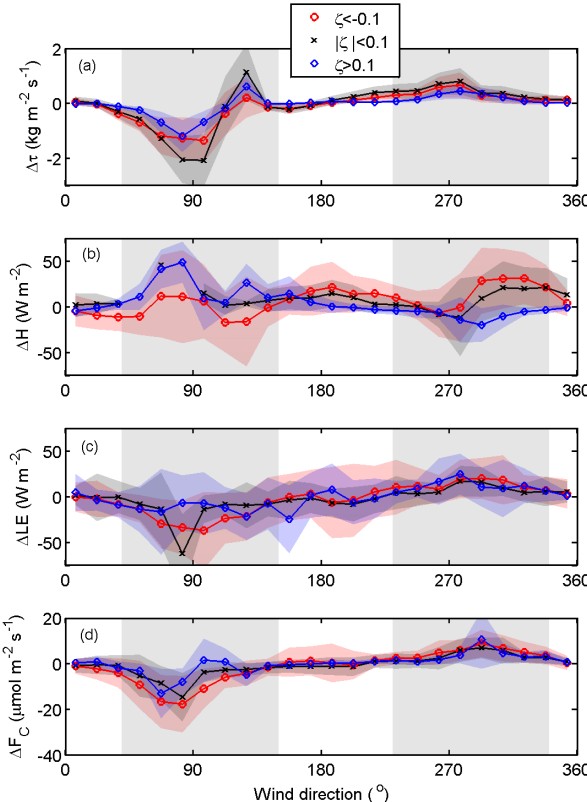

**Figure 4.** Wind direction dependence of the differences in the (a) momentum ($\tau$), (b) sensible ($H$) and (c) latent heat ($LE$), and (d) $CO_2$ ($F_C$) fluxes between the two EC systems (EC1–EC2). Differences are calculated for the whole measurement period and data are separated into different stability classes (unstable ($\zeta < -0.1$), stable ($\zeta > 0.1$) and neutral ($|\zeta| < 0.1$) based on the stability parameter $\zeta$. Lines and symbols represent the $15°$ bin averages and the shaded areas $\pm 1$ std. The neglected wind directions ($40$–$150°$ for EC1 and $230$–$340°$ for EC2) are marked with grey areas.




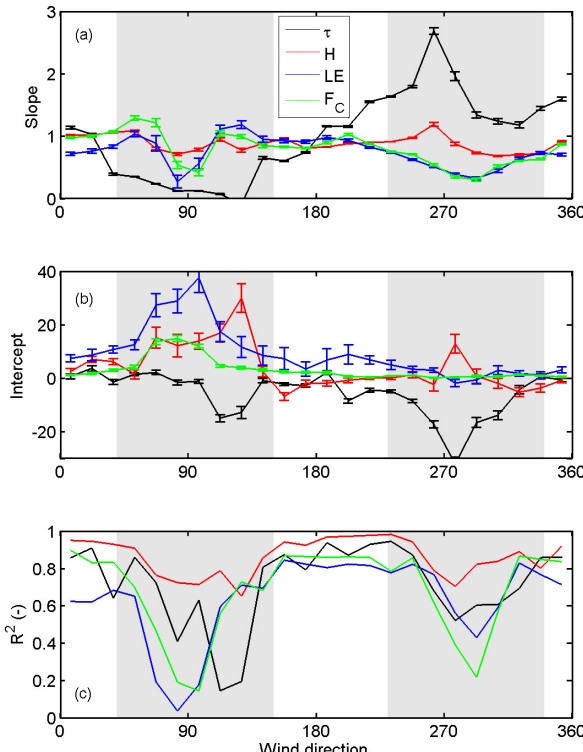

**Figure 5.** Wind direction dependence of the (a) slope, (b) intercept (kg m$^{-2}$ m$^{-1}$, W m$^{-2}$, $\mu$mol m$^{-2}$ s$^{-1}$) and (c) squared correlations coefficient ($R^2$) of the linear least square fit of momentum ($\tau$), sensible ($H$) and latent heat ($LE$), and $CO_2$ ($F_C$) fluxes between the two EC systems (EC2 = Slope·EC1 + Intercept) during July 2013 until September 2015. The neglected wind directions (40–150° for EC1 and 230–340° for EC2) are marked with grey areas.





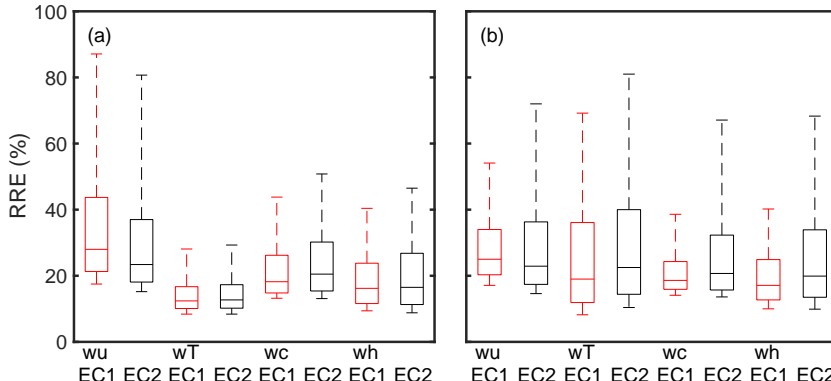

**Figure 6.** Relative random error (RRE) for (a) daytime (solar elevation angle $> 0°$) and (b) night-time (solar elevation angle $\leq 0°$) momentum ($wu$), heat ($wT$), $CO_2$ ($wc$) and water vapour ($wh$) covariances from the two systems EC1 and EC2 outside the flow distortion sectors. Whiskers and boxes represent the 10th, 25th, 50th, 75th and 90th percentiles.

## 3.2 Skewness and kurtosis

$SK$ is within the good data quality limits (-2<$SK$<2) for all studied variables, excluding $CO_2$ (Figure 7, Table 1). Particularly elevated values in the skewness of $CO_2$ ($SK_C$) are seen during the daytime in direction 150–200° with the median $SK_C$ reaching 4 whereas in other directions the medians are around 1. The $90^{th}$ percentiles can reach as high as 5 in direction 150–

200° as it is summarised in Table 1. A similar elevated pattern can also be seen in the kurtosis of $CO_2$ ($K_C$) with the median values reaching 25 indicating spiky behaviour in $CO_2$ (Figure A1). These elevated values are only seen during the daytime so these must relate to the daily activities emitting $CO_2$ and/or prevailing meteorological conditions. The same can clearly be seen from the diurnal variability of both $SK_C$ and $K_C$ shown in Figure 8 for summer months from June till August. While for directions 150–200° elevated values for both statistical variables are seen, in other directions the diurnal variability of $SK_C$

and $K_C$ is relatively flat with the 90th percentiles remaining mostly below 2 and 6, respectively.

   In the direction of elevated $SK_C$ and $K_C$, both variables start to increase in the morning at 6:00 matching with the increase both in road traffic and atmospheric instability observed in Helsinki (Kurppa et al., 2015). Two clear peaks in $SK_C$ and $K_C$ are seen around noon and afternoon between 15:00-19:00. The first peak matches with maxima mixing conditions and the latter with afternoon rush hour. Commonly, at the time of morning rush hour (7:00-9:00) the atmospheric mixing is still relatively

weak and pollutants from the street level are not necessarily as easily transported to the measurement level (Contini et al., 2012; Kurppa et al., 2015). Previously, a skewed distribution of turbulent velocity components within and just above the street canyon has been related to street canyon vortexes causing sweeps and ejections (Oikawa and Meng, 1995). This could also be a potential explanation for the high $SK_C$ and $K_C$ values in direction 150–200° since these directions matches with wind



blowing perpendicular to the streets in the grid type street network in Helsinki. Also previous studies utilising large eddy simulation have shown how street canyon ventilation and sweeps increase in more unstable conditions (Gronemeier et al., 2017; Raupach et al., 2015) which is systematic with our results related to the timing of the maximum $SK_C$ and $K_C$. But the effect of meteorological background conditions cannot be ruled out since the directions with elevated $SK_C$ and $K_C$ match with

flow coming from the sea which can further modify the flow and skewed distribution of $CO_2$ concentration. High skewness values of $CO_2$ data have previously been connected to local-scale anthropogenic sources (Kotthaus and Grimmond, 2012). At the hotel building, small ventilation units are located 9 m below the measurement systems in the north-eastern, north-western and south western corners, but as these do not match the directions 150–200° and systematic signals are seen in both EC1 and EC2, these units cannot be responsible for the increased $SK_C$ and $K_C$. Furthermore, these local-scale sources have been

connected to increased fluxes $F_C$ and $H$ as well as ldecreased $LE$ whereas in our case slightly higher flux values are only seen in $F_C$ in unstable conditions in directions 150–200° (Figure B1). Nonetheless the reason for the elevated $SK_C$ and $K_C$, filtering $F_C$ data based on these variables would remove realistic flux values and therefore they should be used with caution in post-processing of $CO_2$ fluxes.

At the same time with increased $SK_C$ and $K_C$ in the southern direction, the flux stationarity of $F_C$ remains below 0.2 which

is considered to be of high-quality flux data (Figure 9). Thus, applying only the stationarity criteria either with 30 or 60% limit but no skewness and kurtosis criteria would leave most of the data for further data analysis. The most non-stationary variable is the latent heat flux with $90^{th}$ percentiles systematically over 1 in all directions and hours as measured by both setups. $FS_h$ gets slightly greater values with EC1 than EC2 with the first having median values of 0.24 ($90^{th}$ percentile 1.24) in summer and 0.39 (1.56) in winter, and the latter 0.21 (1.08) and 0.39(1.53), respectively. Interestingly, relatively large flux stationary

values of momentum flux are seen both by day and night. Usually, the momentum flux is least filtered based on the stationarity criteria but in our case, due to the complex measurements location, relative large data proportions would be filtered away. The median values are 0.27 (0.69) in summer and 0.17 (0.51) in winter for EC1 being fairly similar to EC2 with median values of 0.28 (0.67) and 0.19 (0.45). Despite the similar magnitude quartile values, EC1 gets hgreater values in directions 190–360° and EC2 symmetrically in directions 0–180°.



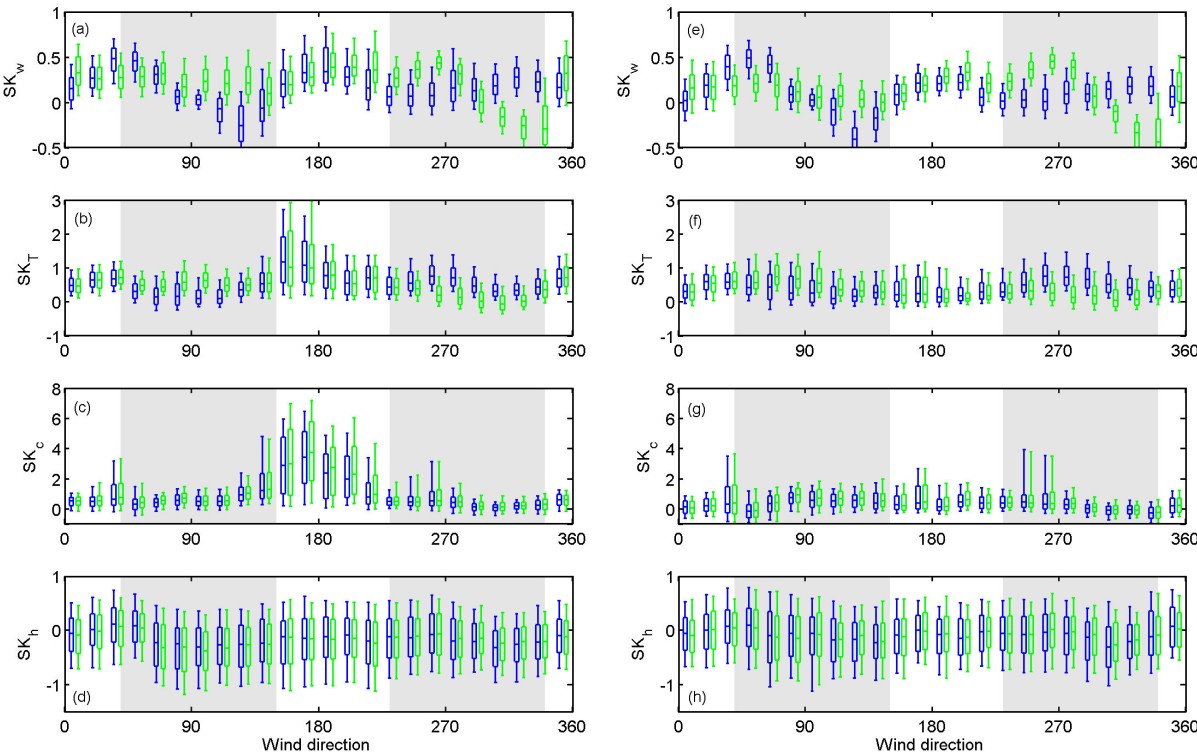

**Figure 7.** Skewness ($SK$) of (a,e) vertical wind speed ($w$), (b,f) air temperature ($T$), (c,g) $CO_2$ ($c$) and (d,h) water vapour ($h$) as a function of wind direction for hours 6:00–21:00 (a-d) and 21:00–6:00 (e-h) for EC1 (blue) and EC2 (green) during July 2013 until September 2015. Whiskers and boxes represent the 10th, 25th, 50th, 75th and 90th percentiles.





**Table 1.** Medians and percentile values (10th, 50th and 90th) of skewness ($SK$), kurtosis ($K$) and flux stationarity ($SF$) of vertical wind speed ($w$), air temperature ($T$), CO$_2$ ($c$) and water vapour ($h$) measured by the two EC setups (EC1 and EC2). Data are separated to summer (June–August) and winter (December–February) and CO$_2$ statistics are differentiated for wind sectors (WD1 150–200 and WD2 the remaining sector). $N$ is the number of data points.

| EC1 | Season | | $N$ | | $SK$ | | | $K$ | | | $FS$ | |
|---|---|---|---|---|---|---|---|---|---|---|---|---|
| $w$ | Summer | | 10335 | -0.09 | 0.17 | 0.56 | 3.1 | 3.5 | 4.4 | 0.06 | 0.27 | 0.69 |
| | Winter | | 8042 | -0.13 | 0.12 | 0.42 | 3.1 | 3.5 | 4.2 | 0.03 | 0.17 | 0.51 |
| $T$ | Summer | | 10333 | 0.06 | 0.55 | 1.25 | 2.7 | 3.6 | 6.0 | 0.04 | 0.18 | 0.68 |
| | Winter | | 8028 | 0.01 | 0.47 | 1.37 | 3.0 | 4.1 | 7.7 | 0.04 | 0.20 | 0.92 |
| $c$ | Summer | WD1 | 633 | -0.02 | 2.07 | 5.15 | 3.4 | 11.7 | 45.0 | 0.02 | 0.12 | 0.45 |
| | | WD2 | 6695 | -0.45 | 0.39 | 1.45 | 2.6 | 3.5 | 9.0 | 0.01 | 0.09 | 0.42 |
| | Winter | WD1 | 967 | -0.03 | 1.54 | 5.73 | 2.8 | 8.4 | 50.3 | 0.01 | 0.04 | 0.24 |
| | | WD2 | 4447 | -0.20 | 0.49 | 2.93 | 2.5 | 3.5 | 22.4 | 0.01 | 0.07 | 0.35 |
| $h$ | Summer | | 8209 | -1.08 | -0.31 | 0.44 | 2.2 | 3.1 | 5.3 | 0.03 | 0.24 | 1.24 |
| | Winter | | 5397 | -0.51 | 0.06 | 0.62 | 2.0 | 2.6 | 3.7 | 0.05 | 0.39 | 1.56 |
| EC2 | Season | | $N$ | | $SK$ | | | $K$ | | | $FS$ | |
| $w$ | Summer | | 10480 | -0.17 | 0.26 | 0.61 | 3.2 | 3.9 | 5.1 | 0.05 | 0.28 | 0.67 |
| | Winter | | 7702 | 0.00 | 0.26 | 0.49 | 3.2 | 3.8 | 4.6 | 0.03 | 0.19 | 0.45 |
| $T$ | Summer | | 10470 | 0.00 | 0.52 | 1.17 | 2.6 | 3.6 | 5.9 | 0.04 | 0.17 | 0.75 |
| | Winter | | 7701 | -0.03 | 0.35 | 1.26 | 3.0 | 4.3 | 8.9 | 0.03 | 0.21 | 1.00 |
| $c$ | Summer | WD1 | 767 | -0.02 | 2.11 | 5.46 | 3.3 | 12.1 | 48.1 | 0.01 | 0.09 | 0.38 |
| | | WD2 | 7617 | -0.44 | 0.43 | 1.67 | 2.6 | 3.6 | 11.0 | 0.01 | 0.07 | 0.36 |
| | Winter | WD1 | 1346 | -0.08 | 1.73 | 6.23 | 2.7 | 9.2 | 60.2 | 0.00 | 0.03 | 0.13 |
| | | WD2 | 6294 | -0.14 | 0.53 | 3.31 | 2.6 | 3.6 | 25.0 | 0.00 | 0.06 | 0.31 |
| $h$ | Summer | | 8232 | -1.10 | -0.32 | 0.40 | 2.3 | 3.1 | 5.3 | 0.03 | 0.21 | 1.08 |
| | Winter | | 7593 | -0.50 | 0.06 | 0.61 | 1.9 | 2.6 | 3.6 | 0.05 | 0.39 | 1.53 |





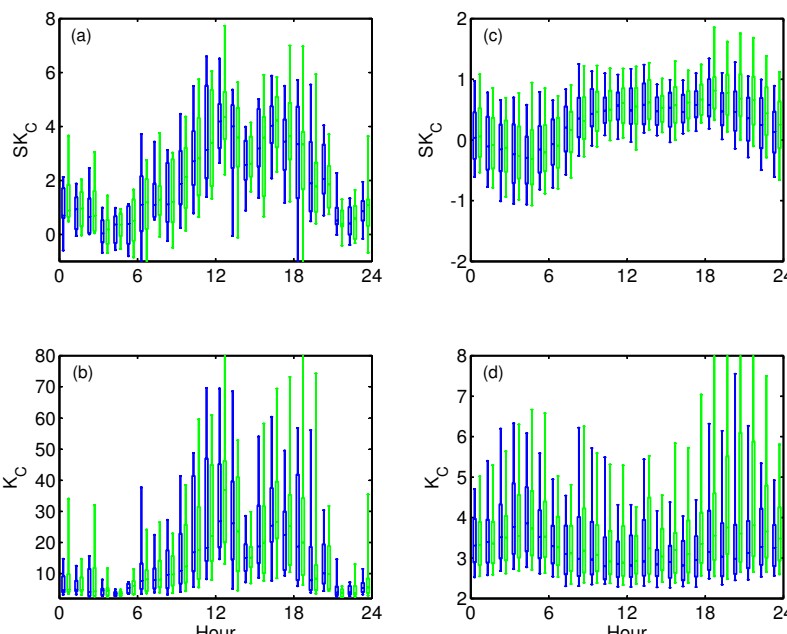

**Figure 8.** Diurnal variability of skewness ($SK$) and kurtosis ($K$) of $CO_2$ for the 150–200° sector (a-b) and for the other directions (c-d) in summer (June to August). Notice the different y-axes on each plot. Whiskers and boxes represent the 10th, 25th, 50th, 75th and 90th percentiles.





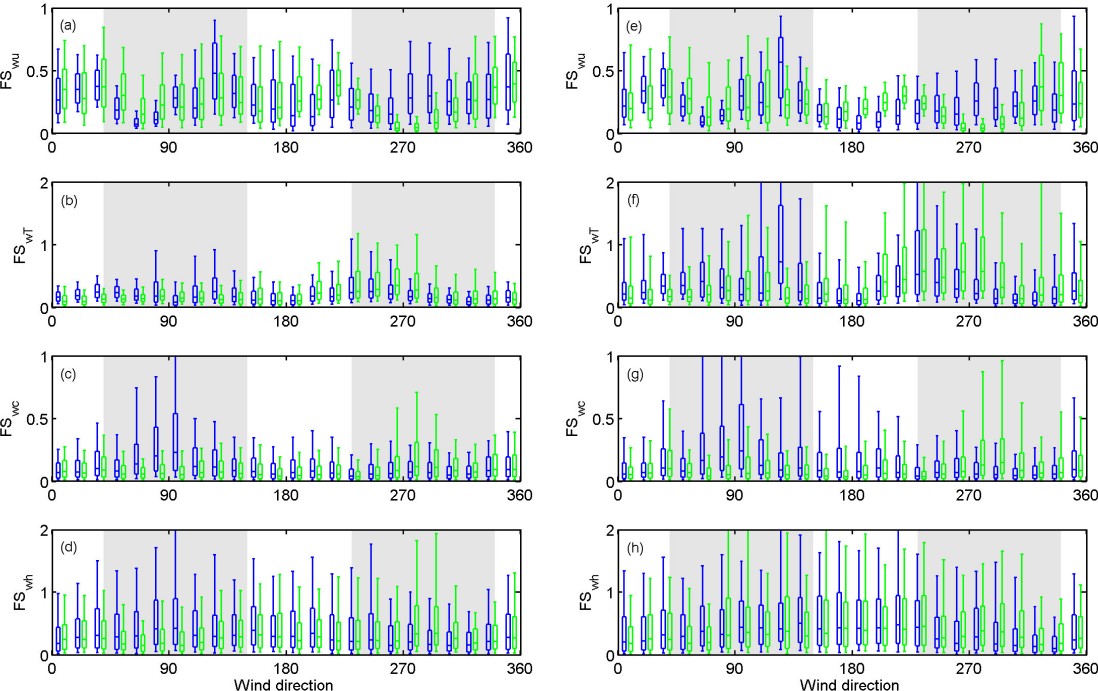

**Figure 9.** Stationarity ($FS$) of (a,e) vertical wind speed ($w$), (b,f) air temperature ($T$), (c,g) $CO_2$ ($c$) and (d,h) water vapour ($h$) as a function of wind direction for hours 6:00–21:00 (a-d) and 21:00–6:00 (e-h) for EC1 (blue) and EC2 (green) during July 2013 until September 2015. Whiskers and boxes represent the 10th, 25th, 50th, 75th and 90th percentiles.

## 3.3   Atmospheric spectra

More information about the similarity/dissimilarity of the two EC systems can be obtained via spectral analysis (Figure 10). The largest differences outside the flow distortion areas can be seen in the co-spectrum of momentum flux with similar contribution only at $n = 0.02 - 0.1$ between the two systems (Figure 10a). With EC1, more contribution is seen at larger eddies and at the

5    inertial subrange the decay is faster than with EC2. A possible explanation for the higher-energy larger eddies is the building wake-effect. With both systems, negative contributions to the total flux are seen at normalised frequencies $> 0.5$ which are likely to be related to the measurement location being on top of a tower. This is supporting the previous findings that velocity components are more impacted by the measurement location than the scalars. Similarly to $\tau$, in the co-spectra of $F_C$ the larger eddies (below normalised frequency 0.03) have slightly more contribution to the total flux measured by EC1 than EC2 and the

10    energy decaying at the inertial subrange ($n > 2$) is faster than in the case of EC2 (Figure 10c). Thus, the flux differences seen



in $\tau$ and $F_C$ between the two systems are to a large extent caused by the larger eddies rather than small-scale variations. For the temperature flux co-variance (Figure 10e), such differences are not seen but rather the contribution of different-sized eddies is very similar between the two systems. Atmospheric spectra of all quantities measured by both systems are similar (Figure 10b,d,f). This indicates different transport mechanisms for temperature and $CO_2$ which has also been found when comparing the transfer efficiencies of the different scalars in this study and in Nordbo et al. (2013) at the same site.

## 3.4 Cumulative surface exchanges

One of the key questions of this study is on how representative a single EC measurement point, in measuring vertical fluxes, can be when the measurements are forced to be conducted close to urban structures causing potentially a large removal of data due to flow distortion areas. After flow distortion and stationarity filtering, the temporal annual coverages at the continuous measurement site EC1 vary from 24–50%, with $H$ and $F_C$ having mean data coverages of 44 % and 45 % when compared to $LE$ 31% (Table 2). The inclusion of the second EC system increases the data coverage substantially with $H$ having mean coverage 65 %, $LE$ 45% and $F_C$ 69 %. The next step is to examine the impact of the different data coverages on the cumulative flux values.

The annual cumulative flux values of $CO_2$, sensible and latent heat calculated for two annual periods (July 2013–June 2014 and July 2014–June 2015) using different gap-filling methods are shown in Figure 11. EC1 and EC2 are gap-filled with their own median cycles using a three-month period around the month being gap-filled with a separation into workdays and weekends. EC1 + EC2 is a combination of EC1 and EC2 systems with data from the first taken in directions 250–330° and the latter in directions 50–130° and in other directions the mean of the two systems is calculated. Missing data were furthermore gap-filled in similar fashion as EC1 and EC2. In the case of $F_C$, EC1+EC2 gives 3–12 % larger cumulative flux values than using only EC1 or EC2 with an annual mean value of 0.375 kmol m$^{-2}$ (Table 2). This indicates that the resulting error in cumulative carbon fluxes due to the single EC measurement point is up to 12 % when other error sources are ignored. For $H$ and $LE$, the differences between the combination data set and EC1 and EC2 are up to 5.3 % and 8.1 %, respectively, with larger cumulative values obtained with EC1 + EC2 than the separate instruments. The difference in $F_C$ is of the same order of magnitude as what has been observed above a forest site within a separation of 30 m between two EC systems (Rannik et al., 2006).

If in addition to the flux stationarity, we would have used the common limits of $K$<1 and $K$>8 and $|SK|$>2 to filter out data, the data coverages of the single EC systems would decrease by 11 % for $F_C$ and 3 and 1 % for $H$ and $LE$ (Table 2). This would give a mean cumulative $F_C$ of 0.3445 kmol m$^{-2}$, which is 3.5 % lower than what obtained by using combination EC1 + EC2 (0.357 kmol m$^{-2}$). Thus, using all $FS$, $SK$ and $K$ to filter our flux data will cause 4.5 % lower cumulative $F_C$ than that of using only $FS$.





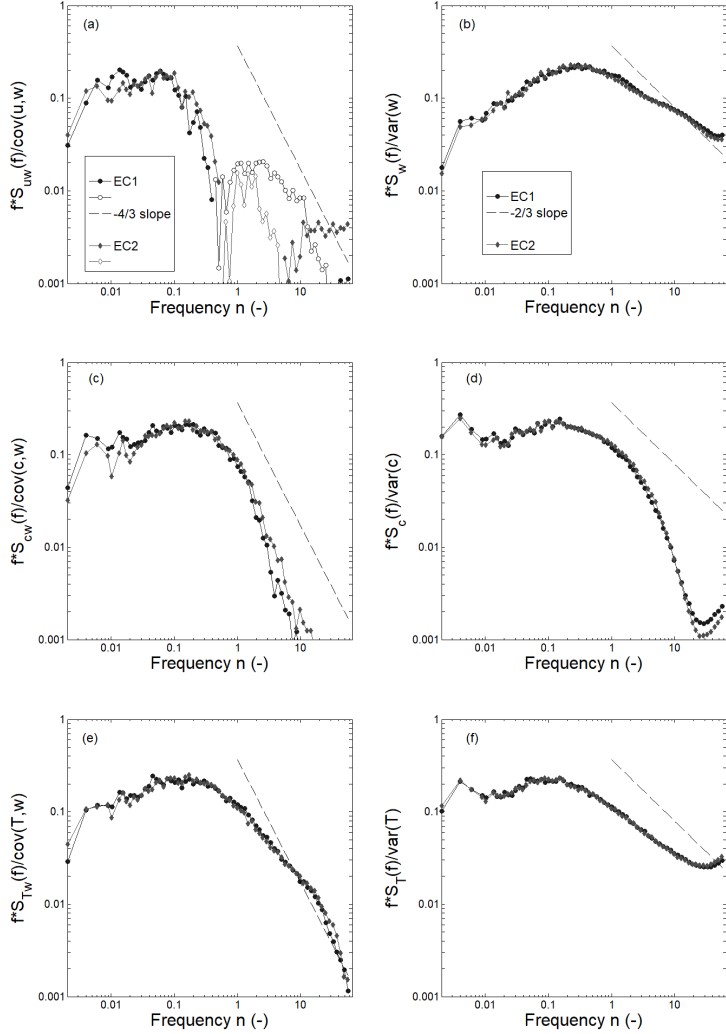

**Figure 10.** Co-spectra (a,c,e) and spectra (b,d,e) of wind ($u$ and $w$-component, respectively), $CO_2$ concentration and air temperature ($T$) as measured with the two EC systems for the undisturbed wind directions for July 2014 ($2.5 < U < 4$ m s$^{-1}$, solar radiation $> 10$ W m$^{-2}$). Solid symbols indicate positive and open symbols negative contributions of the particular frequency. $n = f(z - -d)/U$ is the normalized frequency.





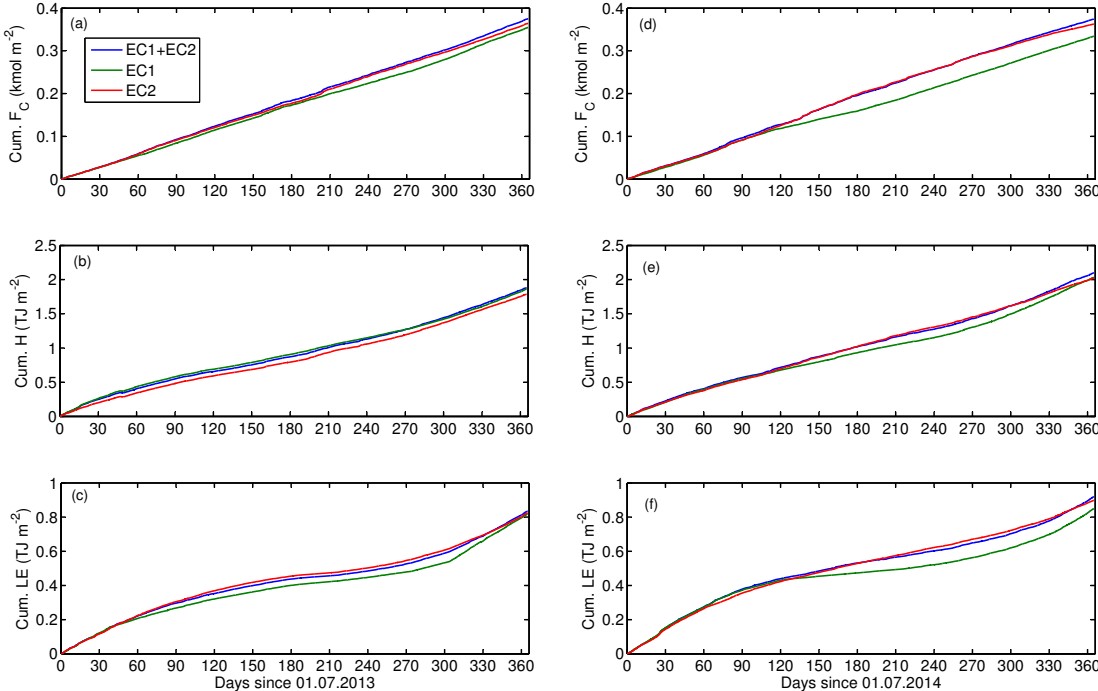

**Figure 11.** Annual cumulative fluxes of (a,b) $CO_2$ ($F_C$), (b,e) sensible ($H$) and (c,f) latent heat ($LE$) for different datasets for July 2013–June 2014 (a-c) and July 2014–June 2015 (d-f). 1$^{st}$ of July. EC1 + EC1 consists of EC1 measurements for sector 230–340°, EC2 measurements for sector 40–150° and the average of the two systems outside the flow distortion sectors (40–150° for EC1 and 230–340° for EC2). The gap-filling of each time series is done based on the diurnal variations over three months period around the month, working days being gapped separately from weekends and holidays.



**Table 2.** Gap-filled cumulative (cum) vertical flux values and percentage of data (%) being gap-filled for two separate years. Fluxes are either filtered using only stationarity ($SF < 0.3$) or stationarity, kurtosis ($K<1$ and $K>8$) and skewness ($|SK|>0.2$). $F_C$ = CO$_2$ flux, $H$ = sensible heat flux and $LE$ = latent heat flux. See Figure 11 caption for details for EC1 + EC2, EC1 and EC2.

| Period | Flux | Filtering | EC1 + EC2 | | EC1 | | EC2 | |
|---|---|---|---|---|---|---|---|---|
| | | | cum | (%) | cum | (%) | cum | (%) |
| | $F_C$ (kmol m$^{-2}$) | | 0.375 | 33.0 | 0.355 | 60.1 | 0.364 | 50.2 |
| $7/2013 - 6/2014$ | $H$ (TJ m$^{-2}$) | $SF$ | 1.880 | 37.4 | 1.861 | 55.6 | 1.786 | 59.2 |
| | $LE$ (TJ m$^{-2}$) | | 0.835 | 56.1 | 0.819 | 72.9 | 0.824 | 73.6 |
| | $F_C$ (kmol m$^{-2}$) | | 0.374 | 29.8 | 0.334 | 59.1 | 0.363 | 51.4 |
| $7/2014 - 6/2015$ | $H$ (TJ m$^{-2}$) | $SF$ | 2.100 | 32.5 | 2.033 | 54.1 | 2.024 | 54.3 |
| | $LE$ (TJ m$^{-2}$) | | 0.919 | 54.3 | 0.850 | 75.6 | 0.898 | 69.9 |
| | $F_C$ (kmol m$^{-2}$) | | 0.357 | 47.1 | 0.343 | 68.8 | 0.346 | 62.8 |
| $7/2013 - 6/2014$ | $H$ (TJ m$^{-2}$) | $SF, K, SK$ | 1.918 | 41.6 | 1.897 | 59.7 | 1.862 | 62.9 |
| | $LE$ (TJ m$^{-2}$) | | 0.827 | 57.2 | 0.814 | 73.5 | 0.816 | 74.3 |
| | $F_C$ (kmol m$^{-2}$) | | 0.367 | 44.5 | 0.320 | 68.1 | 0.365 | 64.8 |
| $7/2014 - 6/2015$ | $H$ (TJ m$^{-2}$) | $SF, K, SK$ | 2.127 | 35.2 | 2.058 | 56.7 | 2.082 | 56.3 |
| | $LE$ (TJ m$^{-2}$) | | 0.913 | 54.9 | 0.839 | 76.0 | 0.896 | 70.3 |

## 4 Conclusions

In this study, simultaneous measurements from two EC systems were compared over highly built-up Helsinki city centre. The identical systems were located symmetrically either side of atop a tower structure with building masonry located in between. Data were identically analysed. This allowed us to examine the sensitivity of a single-point EC system in measuring the vertical

fluxes of momentum, sensible and latent heat, and carbon dioxide, and to understand what are the implications of the non-ideal measurement location and resulting data removal of the studied fluxes.

The flow distortion areas (40–150° and 230–340°) of the two EC systems caused by the building masonry are most clearly distinguishable from wind-normalised TKE. These areas together with a stationarity limit of 30 % resulted in data coverage ranging in 24–50 % with a single system. Outside the flow distortion areas, momentum flux is the most sensitive of all fluxes

for the measurement location and flow modifications caused by the masonry with random uncertainties being around 25 %. With scalar fluxes these remained between 18 and 22 %. Most of the differences in momentum fluxes are due to larger-scale eddies as revealed by spectral analysis indicating larger-scale structures being responsible for the observed differences between these two fluxes.

The two systems had a separation distance of 10 m indicating both systems measuring virtually the same source area and

15 therefore the differences are considered to be caused by variations in flux fields due to the complex surroundings and measure-





ment platform. Despite the measurement location of the EC systems being non-ideal from the point-of-view of flow distortion, the possible bias caused by a single measurement point is less than 12 % for $CO_2$ flux and less than 5 and 8 % for sensible and latent heat fluxes, respectively. In general, the results show how a single point EC measurement can be representative for flux estimates in Helsinki city centre despite the relatively large flow distortion area removing 27 % of the data. This result

is naturally location-specific, but the same result could be considered to apply also in other dense city centres with relative homogeneous road network and other activities around the measurement tower.

We furthermore show that kurtosis and skewness of concentration measurements, common variables used to flag EC data over vegetated surroundings, are not reasonable measures to filter $CO_2$ flux data in dense urban environment due to the combined effect of temporally varying traffic network, meteorological conditions and characteristics of the upwind source area

causing natural spikiness to the $CO_2$ data. Flux stationarity is not impacted in a similar fashion and is therefore considered to be more suitable for filtering $CO_2$ flux data in urban areas. The usage of all three variables to filter out $CO_2$ flux data will cause an underestimation of 4.5 % to annual cumulative carbon fluxes.

Our results are the first from urban areas to characterise the representativeness of single-point EC flux measurements in a densely built urban environment using a combination of two EC systems located close to each other. The related uncertainties

are of the same order of magnitude as observed above vegetated ecosystems. The obtained values can be used as a rule-of-thumb when evaluating in general the representativeness of urban EC measurements used to estimate direct vehicular and building emissions of greenhouse gases and air pollutants. We point out how particular attention should be paid on the data quality control procedures commonly used above vegetated surfaces.

*Data availability.* Data sets used in the data analysis will be saved to and can be freely downloaded from https://b2share.eudat.eu/





**Appendix A**

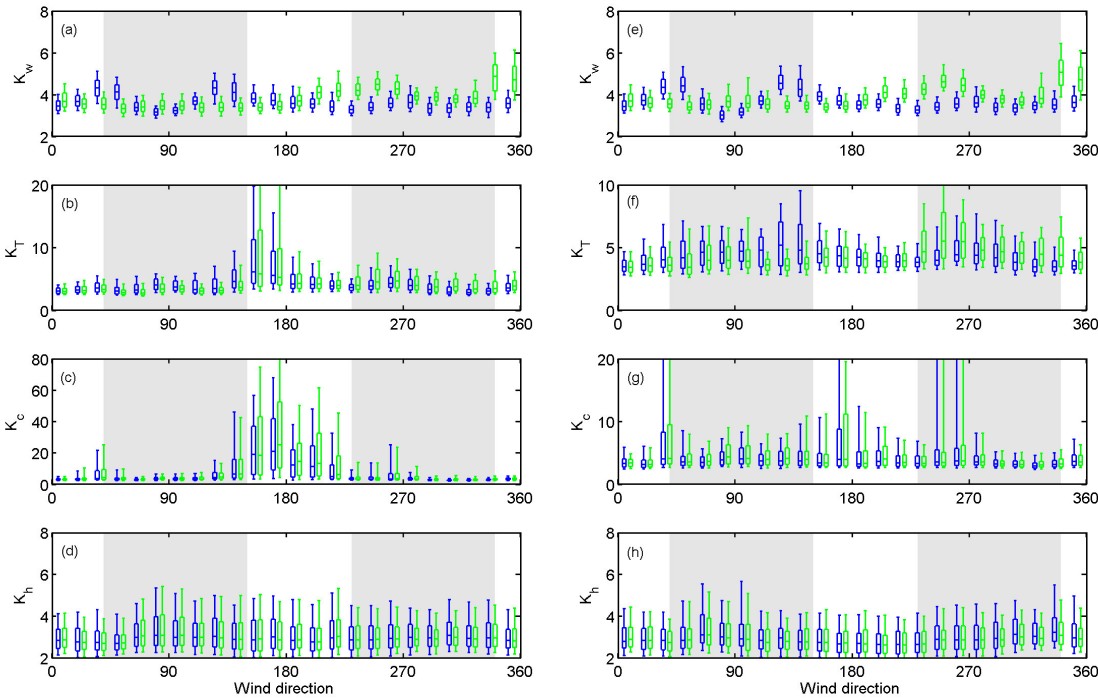

**Figure A1.** Kurtosis ($K$) of (a,e) vertical wind speed ($w$), (b,f) air temperature ($T$), (c,g) $CO_2$ ($c$) and (d,h) water vapour ($w$) as a function of wind direction for hours 6:00–21:00 (a-d) and 21:00–6:00 (e-h) for EC1 (blue) and EC2 (green) during July 2013 until September 2015. Whiskers and boxes represent the 10th, 25th, 50th, 75th and 90th percentiles.

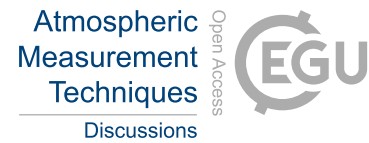

**Appendix B**

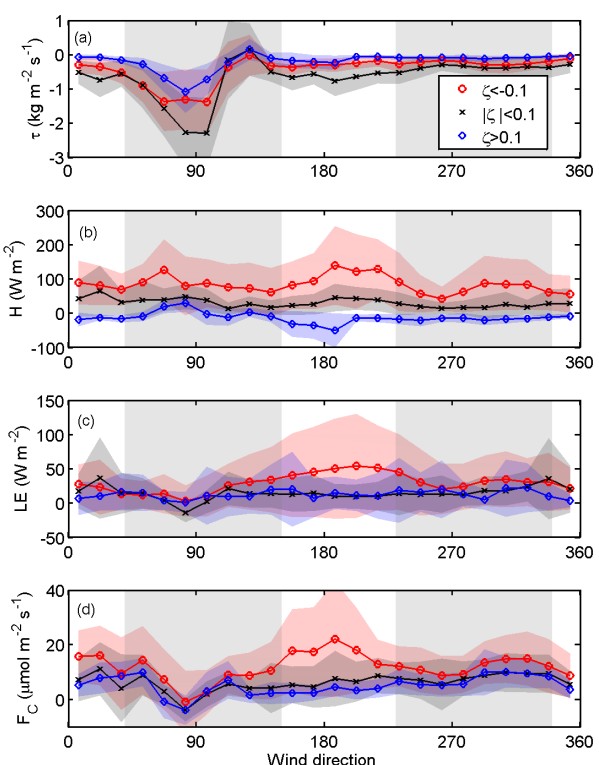

**Figure B1.** Wind-direction dependence of (a) momentum ($\tau$), (b) sensible ($H$) and (c) latent heat ($LE$), and (d) $CO_2$ ($F_C$) fluxes for EC1. The statistics are calculated for the whole measurement period and data are separated into different stability classes (unstable ($\zeta < -0.1$), stable ($\zeta > 0.1$) and neutral ($|\zeta| < 0.1$). Lines and symbols represent the 15° bin averages and the shaded areas ± 1 std. The neglected wind directions (40–150° for EC1 and 230–340° for EC2) are marked with grey areas.

*Author contributions.* L.J., A.K., R.D.K., T.V. and C.R.W planned the measurements; T.V.K. and P.R. were responsible for the eddy covariance measurements; M.K. calculated the eddy covariance data; L.J. and Ü.R performed further data analysis. All authors participated to writing the manuscript.

5   *Competing interests.* The authors declare no competing financial interests.





*Acknowledgements.* The work was supported by the Academy of Finland project ICOS-Finland and Center of Excellence programme (grant no. 307331), and Atmospheric Mathematics collaboration (AtMath) of the Faculty of Science, University of Helsinki, and Maa- ja vesitekni-ikan tuki ry (grant no. 36663). We also thank Sokos Hotel Torni for allowing us to use their building for our EC measurements and Jaakko Kukkonen, Annika Nordbo and Risto Taipale for additional help with the measurements and data analysis.





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
