# Peer review of "Uncertainty of eddy covariance flux measurements over an urban area based on two towers"

_Atmospheric Measurement Techniques, 2018_

## Referee Comment (RC1) · Anonymous Referee #1 · 7 Jun 2018

Manuscript AMT-2018-89 Uncertainty of eddy covariance flux measurements over an urban area based on two towers Järvi et al. General comments: This paper presents a comparison analysis between two identical EC systems in central Helsinki to understand uncertainty of a single point EC measurement of the cumulative vertical fluxes of momentum, sensible and latent heat, and carbon dioxide in a highly dense urban area using several statistics and variables such as stationarity (FS), skewness (SK) and kurtosis (K), relative random uncertainty (RRE), TKE, turbulent transfer efficiencies ($|ruw|$, $|rwt|$) and power and co-spectra. As the authors stress this is the first study using a combination of two close EC systems conducted in a densely built urban area, this research is a step forward for understanding the impact of complex urban structure on fluxes and provides a useful guideline in general for other similar urban EC mea-

surements. However, there are some aspects that need to be illustrate more clearly to improve the manuscript. Please address the comments below and hopefully reflect them into the revised version.

Specific comments: 1. P5 L25. The RRE calculation equation (Eq. 3) is inconsistent with that in Lenschow et al. 1994. In Eq. 3, the square root seems missing. And variables in Eq. 3 are not expressed clearly. i.e. What's the exact formula for calculating the integral time-scale ($\Gamma$f ) or does it have a relation to the averaging period (30min) or have a specific value? How to calculate the correlation coefficient (rws)? Moreover, the expression (rws) is confusing with that of the turbulent transfer efficiencies in Eq. 5 and 6, are they indeed the same or different?

2. P6. For the calculation of spectra and co-spectra (Eq. 7-9), do they have any citations? What do the variables Sx(f) and Sxw(f) represent and what are their formulas? Why are spectra divided into 76 bins? How to determine the frequency f (HZ)? Could you explain all these aspects more clearly, so that potential readers can better understand this work?

3. P6 L24. It's confusing that the angles outside the flow distortion areas are so small: 5-18o for EC1 and 2-15o for EC1. Why aren't these angles ranges excluding the flow distortion area, i.e. 0-40 and 150-360 for EC1, 0-230 and 340-360 for EC2?

4. P8 L19. For daytime (Fig. 6a), the lowest RRE is sensible heat rather latent heat.

5. P8 L23. What's the possible reason for the contrary results between those previous studies (RRE lowest with momentum flux)?

6. P11 L8. Except the summer SKC and KC shown in Fig. 8, how about other months?

7. P16 L5. What does inertial subrange mean? How to distinguish between negative and positive contributions? What do -4/3 slope and -2/3 slope represent in Fig. 10 and what's the basis to clarify between solid and open circles in Fig. 10a? How about other months except Jult=y 2014?

Technical corrections: P2 L21. Replace "paid on" with "paid to". Also in P21 L17. P3 L26. "the systems are located at 60.3 m", isn't it 60 m? P5 L1. "10 1 min-1", what does the space between 0 and 1 mean? P5 Eq. 4. the prime over v is missing. P7 L12. "correlation coefficient (R2)", I think it's determination coefficient. P12 L10. Typo: ldecreased. P12 L23.Typo: hgreater. P13 Fig. 7. To make it more clear, legend is recommended to add into the figure. Fig. 6 and Fig. 7. It would be better to use the same definition for daytime and nighttime. Either based on solar elevation angle (Fig. 6) or hour ranges (Fig. 7). P17 L17-18. The directions 250-330 and 50-130 are still not changed.

(Please also see the attached pdf)

Please also note the supplement to this comment:
https://www.atmos-meas-tech-discuss.net/amt-2018-89/amt-2018-89-RC1-supplement.pdf

---

## Referee Comment (RC2) · O. Menzer (Referee) · 14 Jul 2018

General comments

The Manuscript AMT-2018-89 by Leena Järvi et al. discusses a paired tower approach to assess the representativeness of measurements of vertical momentum, sensible heat, latent heat and carbon dioxide fluxes with the Eddy Covariance technique in a densely built urban environment. The two identical instrument systems were installed on the same building and only 10m apart, therefore they are virtually sampling the same source area outside of flow distortion angles that need to be excluded. The study is relevant to the scientific community since it will (1) help to better understand the relative and absolute magnitudes of measurement errors over an urban, heterogeneous surface area. Furthermore, the approach also allows to (2) assess the representativeness of measurements at sites equipped with a single Eddy Covariance system, specifically when measurements over flow distortion areas need to be excluded which can result in systematic biases in temporally cumulated flux sums. The study was conducted with micrometeorological and statistical rigor; however, a list of technical corrections need to be addressed before publication in Atmospheric Measurement Techniques. Most of the corrections can easily be fixed by quite simple edits and efforts to make the text slightly more concise, predominantly in the Abstract and the Conclusion sections.

————————————-

Specific comments

(1) Given the extensive experimentation, subsequent processing and analysis of the data, the abstract has to be edited and refined quite substantially. In my view, this will ensure the accessibility of the technically dense study to a wider, perhaps even non-technical audience. Some of the sentences are worded in a way that is too vague, so they could be misinterpreted. I am providing more specific feedback on the abstract in the Technical Corrections section below.

(2) The following result that is stated in the abstract appears to be not stated explicitly anywhere throughout the manuscript, or at least I was not able to find it: "the random uncertainties of the two systems are between 10 and 40 %." I suppose the values are simply "read off" of Fig. 6? If this is the case, it looks to me that really what was deducted here is that the interquartile range of the random uncertainties is between 10% and 40 %? I suggest to either provide the range of average (or median) uncertainties, or to add the statistical significance of the uncertainty assessments (e.g., at the 75% or 95% significance level). If the authors feel that would be too much detail for the abstract, then please incorporate this detail into the text of the manuscript. Otherwise the reader may be left guessing where this quite important result came from. Finally,

[Figure]

the wording of this sentence as it stands also allows for some misinterpretation that the random uncertainties are calculated by looking at both towers together (which is ultimately one key objective of the study, cf. the key word "between"), when really the numbers 10%-40% simply represent the random uncertainties of one system at a time.

(3) One thought I kept pondering about while reading the article was the downstream implication of this study for future experimental EC studies in urban environments. Specifically, the results on the representativeness and sensitivities of the measurements as obtained by the paired tower approach. I.e., is the result conclusive of no confirmation being needed at other locations? Are the results truly transferable to other urban locations with fairly homogeneous flux source areas as pointed out at the end of the abstract? Or, is further experimental validation needed? May the authors please discuss this in some more detail, perhaps at the end of section 3.4 in a short paragraph.

(4) It is really encouraging to see that the random uncertainties decreased by applying the paired tower approach in an urban environment. Even better, the relative magnitude of the uncertainties appears to be in the same range as reported by previous studies with more homogeneous terrain. To me, this is one of the key results of the study, and could perhaps even be highlighted in the Abstract since it is highly relevant to future studies conducted in urban or other heterogenous terrain without "directional deviations" in the source area.

——————————————-

Technical Corrections

Abstract, LL3-4: "Often one ecosystem is monitored using only a single EC measurement station bringing uncertainties to the ecosystem-level flux values." I would re-write this to: "Typically an ecosystem is monitored by only one single EC measurement station at a time, making the ecosystem-level flux values subject to random and systematic uncertainties"

Abstract, LL12-14: are "measurement location" and "measurement structures" used synonymously in this sentence? Might not be clear to a wider audience.

Abstract, L18: I suggest writing: "Combining the data from two EC systems also increases the percentage of usable half-hourly carbon fluxes from 45% to 69% at the annual level."

Abstract, LL17-19: I suggest to also give absolute values for the underestimation in grams of Carbon p.a., next to the 12% and 5-8%.

Abstract, L22: Which uncertainties are you referencing here? Random, systematic, or both? Please specify. (If I understood correctly, you are referencing both systematic uncertainties due to excluding flow-distorted wind sectors, and, random uncertainties due to turbulent sampling errors as assessed by the relative random uncertainty (RRE) metric.)

Abstract, L22: I suggest changing "The same results can be assumed to apply in similar dense city locations [. . .]" to "Comparable results can be expected in similarly dense city locations [. . .]"

Pg2, L26: please add a reference.

Pg2, L27: "to reject large amount of data": I suggest writing "to reject a relatively large fraction of the data".

Pg3, L2: I suggest writing "On top of that, any statistical gap-filling technique can be biased [. . .]" instead of "Either way, statistical gap-filling techniques can be biased [. . .]"

Pg3, L8: I would add the study of Hollinger & Richardson (2005) to this list of paired tower approaches, since it was the first of its kind.

Pg3, L19: Figure reference is missing. (???)

Section 2.1: Can the authors please add one sentence in section 2.1 (Site description) on the representativeness of the flux source area as surveyed by the tower with

respect to the "Helsinki city centre" that is referenced further down in the Conclusions (Pg21, L4)? Perhaps simply by referencing information in the original citation for this site. Since the results of the study at hand are discussing the "representativeness" of measurements in a sampling sense, it may be helpful to the reader to be able to put things into perspective. It would also illustrate how essential it is to understand the fluxes extremely well at a fine spatial scale, to then use these measurements as the basis for accurate assessments of larger neighborhood or city level scales.

Pg5, L26: Spelling mistake in the word "square", please run a spell check before submitting the revision.

Pg8, L1: "the median R2 between the two measurement systems is 0.85"

Pg8, L25: this discussion may be more meaningful if an equation for R2 was provided. There are different equations for R2 in the statistical literature. Also, I suggest to rewrite this same sentence and the following to: "Both statistical variables RRE and R2 should theoretically be a measure of random uncertainty. When RRE between the two systems are larger, R2 is expected to be smaller. Furthermore, we expected the two resulting uncertainty rankings (according to RRE and R2) across the different fluxes to be consistent."

Pg12, L10: typo "1decreased"

————————————-

References

Hollinger, D. & Richardson, A. (2005) Uncertainty in eddy covariance measurements and its application to physiological models. Tree Physiology 25, 873-885.

---

## Author Comment (AC1) · 14 Aug 2018

We thank the reviewer for valuable comments and suggestions to improve the manuscript *Uncertainty of eddy covariance flux measurements over an urban area based on two towers.* Please find our point-by-point responses below.

**General comments:**

This paper presents a comparison analysis between two identical EC systems in central Helsinki to understand uncertainty of a single point EC measurement of the cumulative vertical fluxes of momentum, sensible and latent heat, and carbon dioxide in a highly dense urban area using several statistics and variables such as stationarity (FS), skewness (SK) and kurtosis (K), relative random uncertainty (RRE), TKE, turbulent transfer efficiencies (|ruw|, |rwt|) and power and co-spectra. As the authors stress this is the first study using a combination of two close EC systems conducted in a densely built urban area, this research is a step forward for understanding the impact of complex urban structure on fluxes and provides a useful guideline in general for other similar urban EC measurements.

However, there are some aspects that need to be illustrate more clearly to improve the manuscript. Please address the comments below and hopefully reflect them into the revised version.

**Specific comments:**

1. P5 L25. The RRE calculation equation (Eq. 3) is inconsistent with that in Lenschow et al. 1994, which they presented as

Thus, the relative error is

$$\frac{\sigma_F(T)}{|F|} = \left(\frac{2\mathcal{T}_f}{T}\right)^{1/2} \left(\frac{1+r_{ws}^2}{r_{ws}^2}\right)^{1/2}.$$
 (48)

In Eq. 3, the square root seems missing.

We noticed that this form of the relative random error is not directly used in our calculations but rather equation (46) from Lenschow et al. is used, where the flux variance follows the left-hand side of their equation (47). We have now fixed the manuscript accordingly (P5-6, L28-3).

And variables in Eq. 3 are not expressed clearly. i.e. What's the exact formula for calculating the integral time-scale ( $\Gamma f$ ) or does it have a relation to the averaging period (30min) or have a specific value? How to calculate the correlation coefficient (*rws*)? Moreover, the expression (*rws*) is confusing with that of the turbulent transfer efficiencies in Eq. 5 and 6, are they indeed the same or different?

Integral time-scale is defined as the integral over the autocovariance function Rwx and in practice is estimated as the lag when  $R_{wx}$  drops to  $e^{-1}$ . We added this explanation to the manuscript. Its equation follows Lenschow et al. equation (26) and as this is simply an integration of Rwx, we left the equation out.

rws in the old equation (3) is the same as the turbulent transfer coefficient, but as that part of the manuscript was modified, no additional clarification for this connection was made.

We noticed that we use different abbreviation s and x for the variable to which the flux is calculated. To be concise we use x now throughout Section 2.2.

2. P6. For the calculation of spectra and co-spectra (Eq. 7-9), do they have any citations? What do the variables Sx(f) and Sxw(f) represent and what are their formulas? Why are spectra divided into 76 bins? How to determine the frequency f (HZ)?

Could you explain all these aspects more clearly, so that potential readers can better understand this work?

The book by Stull (1988) is probably the best citation on how to calculate spectra and cospectra. We added this to P6, L13. Sx(f) is the power spectra of variable x and Sxw(f) the cospectra between variable x and vertical wind speed. We added a bit more detailed explanation of these to the manuscript (P6, L15-18) but as these are calculated with Fast Fourier Transformation (FFT), which is a commonly known computational method, we decided not to add the detailed methodology on how these are calculated. The methodology can be found from e.g. Stull (1988).

The number of bins over which the raw spectral data are averaged is always a bit arbitrary but 76 bins have been used at the site also in the past (Nordbo et al. 2013). This will not affect the results but rather the visualization of the spectra. *f* is the frequency of the measurements (Hz) and this information was added to the manuscript (P6, L16).

**3. P6 L24. It's confusing that the angles outside the flow distortion areas are so small: 5-180 for EC1 and 2-150 for EC1. Why aren't these angles ranges excluding the flow distortion area, i.e. 0-40 and 150-360 for EC1, 0-230 and 340-360 for EC2?**

By angle here mean the vertical deflection angle and not wind direction. "Vertical deflection" was added to the sentence (P7,L7) to clarify this.

**4. P8 L19. For daytime (Fig. 6a), the lowest RRE is sensible heat rather latent heat.**

Yes. This is true for daytime. We changed the text to "...the lowest to daytime H (medians 12 and 13%)..." (P9, L18).

**5. P8 L23. What's the possible reason for the contrary results between those previous studies (RRE lowest with momentum flux)?**

The reason for the higher RRE in this study is likely the complex measurement location and to emphasize this, we added a sentence "...which is because of the complex measurement location and source-sink distribution at our site." to P9, L23-24.

**6. P11 L8. Except the summer SKC and KC shown in Fig. 8, how about other months?**

Same diurnal behavior is seen on other months but we only added summer here as we would need separate plots for different seasons due to day light saving. We added a sentence "Same behaviour is also seen on other months (not shown)." to the manuscript (P12, L8-9) to be clear.

**7. P16 L5. What does inertial subrange mean? How to distinguish between negative and positive contributions? What do -4/3 slope and -2/3 slope represent in Fig. 10 and what's the basis to clarify between solid and open circles in Fig. 10a? How about other months except July 2014?**

Inertial subrange is the range where turbulent energy is cascading from larger eddies to smaller ones. We added the approximative range of this (n = 0.1-0.4) to the manuscript (P17, L5). We are not sure we understand what the reviewer means with distinguish between negative and positive contributions. These negative values are only observed in the spectra of momentum flux and we clarified this in the manuscript (P17, L6). The value for each normalized frequency is the flux in that frequency bin. The two slopes are those predicted by Kolmogorov's theory. Explanations for these were added to Figure 10 caption: "The -4/3 and -2/3 slopes are those predicted by Kolmogorov". Frequencies with negative contributions are against the net flux and thus we wanted to show how at certain eddy sizes the eddies transport scalars in different directions.

**Technical corrections:** P2 L21. Replace "paid on" with "paid to". Also in P21 L17. Corrected on both locations.**

**P3 L26. "the systems are located at 60.3 m", isn't it 60 m?**

The measurement height is 60.0 m. There was a typo in the text and it has now been fixed.

**P5 L1. "10 1 min-1", what does the space between 0 and 1 mean?**

The units of flow rates is litre per minute and I (not 1) is abbreviation for this. To avoid confusion, we now opened the units.

**P5 Eq. 4. the prime over v is missing.**

Prime added.

**P7 L12. "correlation coefficient (R2)", I think it's determination coefficient.**

Yes. Squared R is the coefficient of determination, which in our case is the square of Pearson correlation coefficient and that is why we used incorrectly only correlation coefficient. We have changed this to coefficient of determination throughout the text.

P12 L10. Typo: Idecreased. Typo fixed.

P12 L23.Typo: hgreater. Typo fixed.

P13 Fig. 7. To make it more clear, legend is recommended to add into the figure. The figure legends were added to Figs. 7, 8 and 9 and A1.

**Fig. 6 and Fig. 7. It would be better to use the same definition for daytime and nighttime. Either based on solar elevation angle (Fig. 6) or hour ranges (Fig. 7).**

We agree that it is a bit unusual to have the different definitions for day and night-time in the two plots, but this was done on purpose due to the need of each figure. In Figure 6 the main point is to show how the RRE varies in different atmospheric conditions, which are to a large extent determined by solar radiation, whereas in Figure 7 we want to refer to human activity which is independent on the different hours of sunlight or darkness. Thus, we prefer leaving the different definitions to the manuscript.

P17 L17-18. The directions 250-330 and 50-130 are still not changed.

The limits have now been corrected also in the main text.

---

## Author Comment (AC2) · 14 Aug 2018

We thank the reviewer for valuable comments and suggestions to improve the manuscript *Uncertainty of eddy covariance flux measurements over an urban area based on two towers*. Please find our point-by-point responses below.

General comments
The Manuscript AMT-2018-89 by Leena Järvi et al. discusses a paired tower approach to assess the representativeness of measurements of vertical momentum, sensible heat, latent heat and carbon dioxide fluxes with the Eddy Covariance technique in a densely built urban environment. The two identical instrument systems were installed on the same building and only 10m apart, therefore they are virtually sampling the same source area outside of flow distortion angles that need to be excluded. The study is relevant to the scientific community since it will (1) help to better understand the relative and absolute magnitudes of measurement errors over an urban, heterogeneous surface area. Furthermore, the approach also allows to (2) assess the representativeness of measurements at sites equipped with a single Eddy Covariance system, specifically when measurements over flow distortion areas need to be excluded which can result in systematic biases in temporally cumulated flux sums. The study was conducted with micrometeorological and statistical rigor; however, a list oftechnical corrections need to be addressed before publication in Atmospheric Measurement Techniques. Most of the corrections can easily be fixed by quite simple edits and efforts to make the text slightly more concise, predominantly in the Abstract and the Conclusion sections.
————————-
Specific comments
(1) Given the extensive experimentation, subsequent processing and analysis of the data, the abstract has to be edited and refined quite substantially. In my view, this will ensure the accessibility of the technically dense study to a wider, perhaps even non-technical audience. Some of the sentences are worded in a way that is too vague, so they could be misinterpreted. I am providing more specific feedback on the abstract in the Technical Corrections section below.

We thank the reviewer for pointing out that the abstract is not necessarily meeting the right audience for the paper. Detailed responses can be found from below.

(2) The following result that is stated in the abstract appears to be not stated explicitly anywhere throughout the manuscript, or at least I was not able to find it: "the random uncertainties of the two systems are between 10 and 40 %." I suppose the values are simply "read off" of Fig. 6? If this is the case, it looks to me that really what was deducted here is that the interquartile range of the random uncertainties is between 10% and 40 %? I suggest to either provide the range of average (or median) uncertainties, or to add the statistical significance of the uncertainty assessments (e.g., at the 75% or 95% significance level). If the authors feel that would be too much detail for the abstract, then please incorporate this detail into the text of the manuscript. Otherwise the reader may be left guessing where this quite important result came from. Finally, the wording of this sentence as it stands also allows for some misinterpretation that the random uncertainties are calculated by looking at both towers together (which is ultimately one key objective of the study, cf. the key word "between"), when really the numbers 10%-40% simply represent the random uncertainties of one system at a time.

Thank you for pointing this out. We agree that it is more meaningful to have here the range of mean values rather than the interquartile range. We changed the range to 12-28% (also reported in the main text). With the suggested modification to the sentence it now reads "the median random uncertainties of the studied fluxes measured by one system are between 12 and 28 %" (P1, L17).

(3) One thought I kept pondering about while reading the article was the downstream implication of this study for future experimental EC studies in urban environments. Specifically, the results on the representativeness and sensitivities of the measurements as obtained by the paired tower approach. I.e., is the result conclusive of noconfirmation being needed at other locations? Are the results truly transferable to other urban locations with fairly homogeneous flux source areas as pointed out at the end of the abstract? Or, is further experimental validation needed? May the authors please discuss this in some more detail, perhaps at the end of section 3.4 in a short paragraph.

The referee raises here a good point. In addition to the abstract this is also shortly mentioned in the conclusions. As suggested we now added further discussion to the end of the Section 3.5 (P20, L25-30):

"The outcome of our study is that a single EC measurement point can produce reasonable estimations for surface fluxes above relatively homogeneous urban surface, but the next question naturally is that how applicable this result is for other urban EC sites. Each urban measurement location is unique and in order to get a final answer, each site should be separately evaluated with more than one measurement setup. Nevertheless, the obtained uncertainties from this study can be used as a first approximation for urban EC measurements in a same way as the few two or multiple tower studies made in vegetated ecosystems are used to give general guidelines for the uncertainties."

(4) It is really encouraging to see that the random uncertainties decreased by applying the paired tower approach in an urban environment. Even better, the relative magnitude of the uncertainties appears to be in the same range as reported by previous studies with more homogeneous terrain. To me, this is one of the key results of the study, and could perhaps even be highlighted in the Abstract since it is highly relevant to future studies conducted in urban or other heterogenous terrain without "directional deviations" in the source area.

We cannot really say that the random uncertainties of the EC observations decrease by applying the two-tower approach as no joint value for the combined dataset is calculated. We only show that the systematic uncertainty decreases and this already mentioned in the abstract. However, we added a sentence "The obtained random and systematic uncertainties are in the same range as observed in vegetated ecosystems." (P1, L21-22) to emphasize the correspondence of the random and systematic uncertainties with those obtained in vegetated ecosystems.

Technical Corrections
Abstract, LL3-4: "Often one ecosystem is monitored using only a single EC measurement station bringing uncertainties to the ecosystem-level flux values." I would re-write this to: "Typically an ecosystem is monitored by only one single EC measurement station at a time, making the ecosystem-level flux values subject to random and systematic uncertainties"

Corrected as suggested.

Abstract, LL12-14: are "measurement location" and "measurement structures" used synonymously in this sentence? Might not be clear to a wider audience.

Yes. To clarify this, we removed measurement structures from the sentence and now it simply reads "The momentum flux is the most sensitive to the measurement location whereas scalar fluxes are less impacted" (P1, L13-14).

Abstract, L18: I suggest writing: "Combining the data from two EC systems also increases the percentage of usable half-hourly carbon fluxes from 45% to 69% at the annual level."

Modified to "Combining the data from two EC systems also increases the fraction of usable half-hourly carbon fluxes from 45 % to 69 % at the annual level." (P1, L19-21)

Abstract, LL17-19: I suggest to also give absolute values for the underestimation in grams of Carbon p.a., next to the 12% and 5-8%.
Added as suggested. We also added the values in units g C m-2 to the manuscript main text.

Abstract, L22: Which uncertainties are you referencing here? Random, systematic, or both? Please specify. (If I understood correctly, you are referencing both systematic uncertainties due to excluding flow-distorted wind sectors, and, random uncertainties due to turbulent sampling errors as assessed by the relative random uncertainty (RRE) metric.)
We refer to systematic uncertainty and this is now clarified also in the sentence (P2, L4).

Abstract, L22: I suggest changing "The same results can be assumed to apply in similar dense city locations [...]" to "Comparable results can be expected in similarly dense city locations [...]"
Corrected as suggested (P2, L4).

Pg2, L26: please add a reference.
Reference added (P2, L31).

Pg2, L27: "to reject large amount of data": I suggest writing "to reject a relatively large fraction of the data".
Corrected as suggested (P2, L31).

Pg3, L2: I suggest writing "On top of that, any statistical gap-filling technique can be biased [...]" instead of "Either way, statistical gap-filling techniques can be biased [...]"
Corrected as suggested (P3, L7).

Pg3, L8: I would add the study of Hollinger & Richardson (2005) to this list of paired tower approaches, since it was the first of its kind.
Added as suggested (P3, L13).

Pg3, L19: Figure reference is missing. (???)
This was fixed.

Section 2.1: Can the authors please add one sentence in section 2.1 (Site description) on the representativeness of the flux source area as surveyed by the tower with respect to the "Helsinki city centre" that is referenced further down in the Conclusions
We added sentence "The two systems have a separation distance of 10 m and thus measure virtually the same source area" to P4, L2-3.

(Pg21, L4)? Perhaps simply by referencing information in the original citation for this site. Since the results of the study at hand are discussing the "representativeness" of measurements in a sampling sense, it may be helpful to the reader to be able to put things into perspective. It would also illustrate how essential it is to understand the fluxes extremely well at a fine spatial scale, to then use these measurements as the basis for accurate assessments of larger neighborhood or city level scales.
We modified the sentence to (P23, L4-8) "This result is naturally location-specific for this highly built-up site with vegetation fraction only 22% and relatively homogeneous roof level

(Nordbo et al. 2013). The same result could be considered to apply also in other dense city centers with similar relatively homogeneous surface characteristics."

Pg5, L26: Spelling mistake in the word "square", please run a spell check before submitting the revision.
Done.

Pg8, L1: "the median R2 between the two measurement systems is 0.85"
Added as suggested (P8, L11).

Pg8, L25: this discussion may be more meaningful if an equation for R2 was provided. There are different equations for R2 in the statistical literature. Also, I suggest to rewrite this same sentence and the following to: "Both statistical variables RRE and R2 should theoretically be a measure of random uncertainty. When RRE between the two systems are larger, R2 is expected to be smaller. Furthermore, we expected the two resulting uncertainty rankings (according to RRE and R2) across the different fluxes to be consistent."
We added sentence "calculated as the square of the Pearson correlation coefficient" to P8, L6. We do not calculate joint RRE for the two setups but rather separate RRE's for the two systems. We modified the change to "Both statistical variables RRE and R2 should theoretically be a measure of random uncertainty. When RREs measured with the two systems are larger, R2 between the two systems is expected to be smaller. Furthermore, we expected the two resulting uncertainty rankings (according to RRE and R2) across the different fluxes to be consistent." (P9, L25-28).

Pg12, L10: typo "1decreased"
Typo fixed.